# FAST THINKING FOR LARGE LANGUAGE MODELS

## ABSTRACT

Reasoning-oriented Large Language Models (LLMs) often rely on generating explicit tokens step by step, and their effectiveness typically hinges on large-scale supervised fine-tuning or reinforcement learning. While Chain-of-Thought (CoT) techniques substantially enhance performance on complex reasoning tasks, they remain inefficient, requiring long reasoning traces that increase latency and token usage. In this work, we introduce *Latent Codebooks for Fast Thinking*, a framework that uses concise CoT sketches only during training to learn a codebook of discrete strategy priors. At inference, the model conditions on a handful of continuous thinking vectors distilled from the codebook in a single pass, enabling strategy-level guidance without producing explicit reasoning tokens. To complement this design, we propose GAINROUTER, a lightweight routing mechanism that adaptively switches between fast codebook-guided inference and slow explicit reasoning, thereby suppressing overthinking and reducing unnecessary token generation. Experiments across multiple reasoning benchmarks show that our approach achieves competitive or superior accuracy while substantially lowering inference cost, offering a practical path toward efficient and controllable reasoning in large language models.

## 1 INTRODUCTION

Large language models (LLMs) Achiam et al. (2023); Dubey et al. (2024); Yang et al. (2025) have achieved strong performance across natural language understanding, code generation, and mathematical reasoning Yang et al. (2024a); Roziere et al. (2023); Kuang et al. (2025). To further enhance the complex reasoning capacity of LLMs, the community widely adopts *Chain-of-Thought* (CoT) and related multi-step prompting strategies, which decompose a task into compositional subgoals and often improved robustness and interpretability for arithmetic, inductive, and long-dependency problems Wei et al. (2022); Kojima et al. (2022). Beyond single trajectories, exploration-based variants such as self-consistency, tree- or graph-structured search, and tool-augmented pipelines further enhance problem solving by sampling or searching over intermediate states Wang et al. (2025a).

Despite their benefits, CoT-style *slow thinking* typically incurs substantial latency and token costs, as the model must emit many intermediate "thinking tokens" to reach an answer Besta et al. (2024); Yang et al. (2024b). On simple instances, expanded reasoning can introduce *overthinking*—unnecessary, loosely relevant steps that may even exacerbate errors Chen et al. (2024). Moreover, strong multi-step reasoning in practice often relies on either *process supervision* or *preference-based policy optimization*—including RL-based methods (e.g., PPO-/GRPO-style RLHF) and RL-free approaches such as DPO—which raises data and engineering barriers for large-scale deployment Schulman et al. (2017); Rafailov et al. (2023); Shao et al. (2024). In this paper, we define the term *slow thinking* to describe these explicit multi-step generation procedures.

Many problems that appear complex at first glance can be solved quickly by quickly recalling and applying prior *solution patterns*, without enumerating long derivations. Analogous to human problem solving—where one "recognizes the situation" early and commits to a high-probability plan—we ask: can an LLM internalize a compact set of reusable, *discrete priors* and, at inference time, fetch a small amount of targeted guidance that *single-pass* conditions decoding? If so, the model could produce high-quality answers via *fast thinking*, avoiding the overhead and potential brittleness of verbose step-by-step generation.

We propose `Latent Codebooks for Fast Thinking` (LC-FT), a fast-thinking–first paradigm that learns a latent *codebook* of reusable priors and conditions decoding with a small set

Figure 1: **Overview of our framework.** Latent Codebook for Fast Thinking (LC-FT) provides hint tokens for efficient one-pass reasoning, while GainRouter decides between fast and slow CoT modes to balance accuracy and token cost.

of *thinking tokens* in a single pass. The codebook serves as strategy-level memory: at test time the model retrieves just enough latent guidance to initiate the solution path, rather than emitting lengthy intermediate text. LC-FT avoids extra process supervision and RL-style policy optimization; it is trained with task-specific data via semantic alignment and standard supervised fine-tuning.

Besides, to automate the choice between fast and slow reasoning, we introduce GAINROUTER, a tiny inference-time gate that combines problem features with codebook-derived uncertainty signals. By default, it keeps the model in fast mode and escalates to CoT reasoning only when the predicted failure risk is high, thus reducing overthinking and token usage while maintaining accuracy. Fig. 1 provides an overview of the framework.

On mathematical reasoning and programming benchmarks, LC-FT attains competitive single-pass accuracy, showing that learned *thinking tokens* effectively guide the solution search. With GAIN-ROUTER, the system preserves near-CoT accuracy yet markedly cuts generated tokens and overthinking, resulting in a controllable accuracy–cost trade-off with minimal training and parameter overhead.

## 2 RELATED WORK

### 2.1 CHAIN-OF-THOUGHT AND PROCESS SUPERVISION

Chain-of-Thought (CoT) reasoning has become central for enhancing multi-step reasoning in LLMs. Prompt-based methods encourage explicit intermediate step generation, improving compositional reasoning on arithmetic, commonsense, and coding tasks (Wei et al., 2022; Kojima et al., 2022; Wang et al., 2022; Zhou et al., 2022). However, these approaches require costly inference-time prompting with multiple sampled chains. To address this limitation, supervised CoT fine-tuning uses annotated or synthetic rationales to train step-by-step reasoning (Lightman et al., 2023), improving consistency but still requiring expensive rationale collection. Building on this, process- or preference-level supervision further optimizes both final outcomes and intermediate step quality (Ouyang et al., 2022; Rafailov et al., 2023; Jiang et al., 2024), reducing error propagation yet needing complex feedback pipelines. Despite these advances, a fundamental trade-off remains: while longer reasoning chains generally improve accuracy (Ahn et al., 2024), they also increase token usage, latency, and overthinking vulnerability. To resolve this tension, recent continuous-space reasoning approaches bypass discrete token decoding (Cheng & Van Durme, 2024; Xu et al., 2025), using continuous hint vectors at inference to better balance reasoning quality and efficiency.

### 2.2 ROUTING FOR REASONING AND EFFICIENCY

Early adaptive computation approaches (Zhu et al., 2024) investigate policies for deciding when CoT reasoning should be triggered based on input complexity. Recent routing strategies extend

this by dynamically determining how language models handle queries based on complexity and uncertainty (Qu et al., 2025; Sui et al., 2025), automatically allocating simple queries to efficient processing while reserving complex ones for powerful reasoning. Current approaches fall into two categories. Inter-model routing methods like MixLLM (Wang et al., 2025b), which enhances query embeddings with tag knowledge and lightweight predictors, and RouteLLM (Ong et al., 2024), which trains routers using Chatbot Arena preference data (Chiang et al., 2024), focus on balancing performance across different models. Intra-model routing approaches take a different direction: Sketch-of-Thought (Aytes et al., 2025) uses lightweight routers to select reasoning paradigms, while Self-Ref (Chuang et al., 2024) enables autonomous routing through intrinsic uncertainty scores. Building on this foundation, our work introduces GAINROUTER, a lightweight gating mechanism that toggles between fast codebook-guided inference and slow explicit CoT to minimize token generation while preserving reasoning quality.

## 3 METHOD

### 3.1 CONCISE COT FOR LATENT PRIOR LEARNING

We build a dataset of *concise* teacher rationales used only as training signals as shown in Fig. 2. Given problem $x$ and gold label $y$, a teacher $\mathcal{T}$ generates a hint $\tilde{r}$. A hint is accepted iff it does not leak $y$ and a student (act) model $f_\theta$ *conditioned* on $\tilde{r}$ predicts $y$ correctly:

$$\text{Valid}(x, \tilde{r}, y) \iff \neg \text{Leak}(\tilde{r}, y) \wedge \text{Verify}\big(f_\theta(\langle x; \tilde{r}\rangle), y\big),$$

where $\langle x; \tilde{r}\rangle$ denotes input conditioning by *concatenating* $\tilde{r}$ *after* $x$ in the input to $f_\theta$, $\text{Leak}(\tilde{r}, y)$ detects canonicalized answer leakage, and $\text{Verify}(\cdot)$ is a task-specific checker. If $\tilde{r}^{(t)}$ fails verification, we re-query $\mathcal{T}$ with the error history $h^{(t)} = \{(\tilde{r}^{(s)}, \hat{y}^{(s)})\}_{s \leq t}$ to obtain $\tilde{r}^{(t+1)}$, and *terminate as soon as* verification succeeds, keeping the first valid hint. The final dataset is $\mathcal{D} = \{(x_i, \tilde{r}_i, y_i)\}_{i=1}^{N}$, enabling *latent prior supervision* without long test-time reasoning.

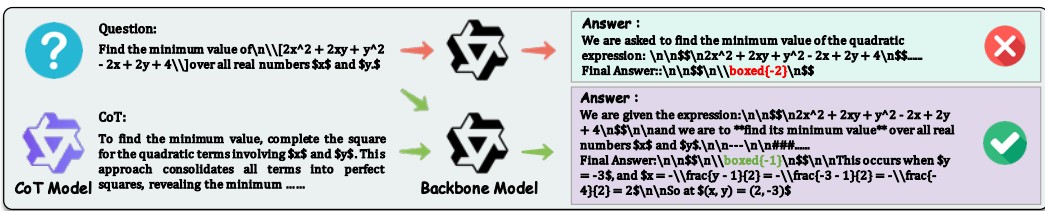

Figure 2: **Pipeline for constructing concise CoT data.** A teacher model generates candidate rationales, a backbone model verifies whether they lead to the correct answer, and the filtered triplets $(x, \tilde{r}, y)$ are collected into the dataset $\mathcal{D}$ for latent prior learning.

### 3.2 LATENT CODEBOOK FOR FAST THINKING

To distill concise rationales into the model, we introduce a latent codebook that stores discrete priors and generates continuous *thinking tokens* to condition decoding as shown in Fig. 3. Let $\mathbf{C} \in \mathbb{R}^{M \times H}$ denote a learnable codebook with $M$ prototype entries, where $H$ is the model hidden size. We use $K$ learnable queries $\mathbf{Q} \in \mathbb{R}^{K \times H}$ *shared across inputs* to attend over the codebook via

$$\mathbf{A} = \text{softmax}\left(\frac{\mathbf{Q}\mathbf{W}_q(\mathbf{C}\mathbf{W}_k)^\top}{\sqrt{H}}\right), \qquad \mathbf{T} = \mathbf{A}(\mathbf{C}\mathbf{W}_v)\mathbf{W}_o \in \mathbb{R}^{K \times H},$$

where $\mathbf{T} = [\mathbf{t}_1; \ldots; \mathbf{t}_K]$ contains $K$ thinking-token vectors $\mathbf{t}_k \in \mathbb{R}^H$, and $M$ is the number of codebook entries. Learnable queries provide a fixed-length interface for variable-length questions, and the attention $\mathbf{A}$ composes priors from multiple prototypes. The codebook serves as a finite basis of reasoning strategies, and $\mathbf{T}$ forms soft compositions of these strategies.

To incorporate the thinking tokens, we reserve $K$ slots in the hidden sequence at transformer layer $L$ (the insert-layer index). Let $\mathbf{Z}^{(L)} \in \mathbb{R}^{(n+K) \times H}$ denote the hidden sequence at layer $L$ with $n$

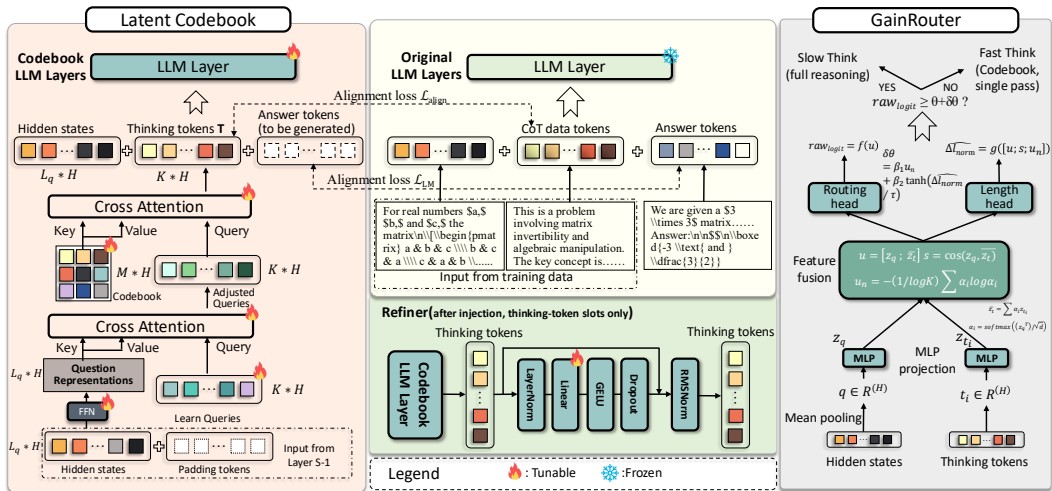

Figure 3: **Overview of the latent codebook mechanism.** Learnable queries $\mathbf{Q}$ attend over the codebook $\mathbf{C}$ to produce *thinking tokens* $\mathbf{T}$, which are refined and injected at layer $L$ to form $\mathbf{Z}^{(L)}$. Thinking tokens are masked before $L$ and visible after $L$, and persist through layers $l > L$ to condition decoding with strategy-level priors.

input tokens and $K$ thinking-token slots. Injection is

$$\mathbf{Z}^{(L)} = \left[ z_1, \ldots, z_n;\ \mathbf{t}_1, \ldots, \mathbf{t}_K \right],$$

where $z_i$ are contextual embeddings and $\mathbf{t}_k$ are the thinking tokens. Thinking tokens are masked before layer $L$ and become visible only after $L$. All subsequent layers $l > L$ then operate on $\mathbf{Z}^{(l)}$, so the thinking tokens influence the remaining computation. A lightweight refiner applies a residual MLP *after* injection and only on the thinking-token slots:

$$\mathrm{Refine}(\mathbf{T}) = \mathbf{T} + \sigma(\mathbf{T}\mathbf{U}_1)\mathbf{U}_2.$$

This introduces strategy-level priors at layer $L$ and does not increase output generation length at inference; the hidden sequence adds only a constant $K$ positions.

### 3.3 TWO-STAGE TRAINING

We train in two stages. **Stage 1 (alignment).** A frozen reference LLM processes the input with the concise rationale appended. At layer $l$ we mean-pool hidden states on the rationale span to obtain $z_R^{(l)}$. Our model injects $K$ thinking tokens at layer $L$; at layer $l$ we mean-pool the thinking-token slots to obtain $z_T^{(l)}$. The alignment loss is

$$\mathcal{L}_{\mathrm{align}}^{(L)} = 1 - \cos\left(z_T^{(L)}, \mathrm{sg}(z_R^{(L)})\right), \qquad \mathcal{L}^{(1)} = \alpha\, \mathcal{L}_{\mathrm{align}}^{(L)} + \beta\, \frac{1}{|S|} \sum_{l \in S} \left(1 - \cos\left(z_T^{(l)}, \mathrm{sg}(z_R^{(l)})\right)\right),$$

where $S = \{L+1, \ldots\}$ and sg stops gradients to the reference model; $\alpha, \beta$ are scalars.

**Stage 2 (supervised fine-tuning).** We drop explicit rationales and train the model to solve tasks conditioned on thinking tokens. For $(x, y)$ we inject the tokens at layer $L$ and optimize

$$\mathcal{L}_{\mathrm{LM}} = -\sum_{t=1}^{|y|} \log p_\theta(y_t \mid y_{<t},\ x,\ \mathbf{T}),$$

updating the base model together with the codebook; adapters are unfrozen from layer $L$ onward.

### 3.4 INSERT-LAYER–AWARE FINE-TUNING WITH LORA

To adapt the backbone efficiently, we apply low-rank adapters Hu et al. (2022) only from layer $L$ onward, where the *thinking tokens* are injected. Earlier layers remain frozen, concentrating adaptation capacity on the sub-network that first interacts with these tokens.

Concretely, for each linear transformation $\mathbf{W} \in \mathbb{R}^{d_{\text{out}} \times d_{\text{in}}}$ in attention or feed-forward blocks with index $l \geq L$, we use

$$\mathbf{W}' = \mathbf{W} + \mathbf{B}\mathbf{A},$$

where $\mathbf{A} \in \mathbb{R}^{r \times d_{\text{in}}}$ and $\mathbf{B} \in \mathbb{R}^{d_{\text{out}} \times r}$ are trainable low-rank factors with $r \ll \min(d_{\text{in}}, d_{\text{out}})$. The base weight $\mathbf{W}$ is kept frozen; only $\mathbf{A}, \mathbf{B}$ are updated.

During **Stage 2** training, we unfreeze adapters from layer $L$ onward and optimize the standard autoregressive objective while keeping earlier layers fixed:

$$\min_{\theta_{\text{codebook}}, \, \theta_{\text{adapters}}, \, l \geq L} \mathcal{L}_{\text{LM}}.$$

This localizes adaptation to the sub-network that consumes the thinking tokens, improving efficiency and stability.

## 3.5 GAINROUTER: PREVENTING OVERTHINKING AND TOKEN WASTE

Latent codebooks enable a fast path, yet some inputs still require explicit multi-step reasoning. **GainRouter** is a lightweight classifier that decides when to switch.

From the non-thinking pass we obtain a pooled question vector $q \in \mathbb{R}^H$ (hidden size $H$) and $K$ thinking-token vectors $\{\mathbf{t}_i\}_{i=1}^K \subset \mathbb{R}^H$. After projection to $\mathbb{R}^d$ (router projection dimension) we get $z_q$ and $\{z_{t_i}\}$. We aggregate token evidence by attention,

$$\bar{z}_t = \sum_{i=1}^K \alpha_i z_{t_i}, \qquad \alpha_i = \text{softmax}_i\left(\frac{z_q^\top z_{t_i}}{\sqrt{d}}\right),$$

form $u = [z_q; \bar{z}_t] \in \mathbb{R}^{2d}$, and map $u$ with a small MLP to $\text{raw}_{\text{logit}}(x)$. Auxiliary features are the cosine $s = \cos(z_q, \bar{z}_t)$ and the normalized *attention* ehntropy

$$u_n = -\frac{1}{\log K} \sum_{i=1}^K \alpha_i \log \alpha_i.$$

We define the normalized length difference

$$\Delta\ell_{\text{norm}} = \frac{\ell_{\text{think}} - \ell_{\text{non}}}{s_\ell},$$

where $s_\ell > 0$ is a fixed length scale (e.g., the mean non-thinking length on the router's training set). A regression head predicts $\widehat{\Delta\ell}_{\text{norm}}$ from $[u; s; u_n]$. The adaptive threshold and decision are

$$\delta\theta(x) = \beta_1 u_n + \beta_2 \tanh\left(\widehat{\Delta\ell}_{\text{norm}}/\tau\right), \qquad \text{USETHINK}(x) = \mathbb{1}\left[\text{raw}_{\text{logit}}(x) \geq \theta + \delta\theta(x)\right],$$

with learnable scalars $\theta, \beta_1, \beta_2$ and temperature $\tau$.

*Training.* Labels come from paired fast/slow executions: positive if only the slow model succeeds; negative if the fast model succeeds (alone or jointly); both-fail cases are ignored. We minimize the class-balanced routing loss

$$\mathcal{L}_{\text{route}} = \text{BCEWithLogits}\left(\text{raw}_{\text{logit}}(x) - (\theta + \delta\theta(x)), \, y(x)\right),$$

with auxiliaries

$$\mathcal{L}_{\text{len}} = \text{SmoothL1}\left(\widehat{\Delta\ell}_{\text{norm}}, \, \Delta\ell_{\text{norm}}\right), \qquad \mathcal{L}_{\text{reg}} = \|\beta\|_2^2,$$

and objective

$$\mathcal{L} = \mathcal{L}_{\text{route}} + \lambda_{\text{len}}\mathcal{L}_{\text{len}} + \lambda_{\text{reg}}\mathcal{L}_{\text{reg}}.$$

GainRouter stays fast by default and triggers slow reasoning only above an adaptive threshold, reducing overthinking and token usage without sacrificing robustness.

## 4 EXPERIMENTS

### 4.1 BENCHMARKS

We evaluate on two mathematical reasoning datasets and two programming benchmarks. For reasoning, we use **AIME 2024/2025** (Balunović et al., 2025) and **OlympiadBench** (He et al., 2024). OlympiadBench is split into **1,832** training examples and **792** test examples; the training split is used to learn the codebook and router, and evaluation is performed on the OlympiadBench test split and the unseen AIME benchmarks. For programming, we train on the official **MBPP** (Austin et al., 2021) training set and evaluate on its test split as well as the full **HumanEval** (Chen et al., 2021).

### 4.2 MODELS

We adopt **Qwen3-4B** Yang et al. (2025) as the base model, which supports both answer-only inference and explicit CoT reasoning. We use `Qwen3-4B-Instruct-2507` as the non-thinking backbone for our codebook-augmented fast thinking model, and `Qwen3-4B-Thinking-2507` as the slow CoT baseline. To construct concise rationales for latent prior learning, we rely on a stronger teacher, **Qwen-Max** Team (2024a). We also extend our framework to `Qwen2-1.5-Instruct` Team (2024b) to study cross-scale distillation.

### 4.3 BASELINES

We compare against four categories of baselines: (i) *prompting* methods (zero-shot and few-shot), (ii) task-adapted fine-tuning with LoRA using concise CoT supervision, (iii) **SoftCoT** (Xu et al., 2025), which injects soft reasoning tokens, (iv) the zero-shot CoT setting of Qwen3-Thinking. More details about the baselines are provided in Appendix A.1.3.

### 4.4 IMPLEMENTATION DETAILS

We use greedy decoding (max 30,720 tokens). We report Pass@1 for programming and exact-match accuracy for reasoning. Hyperparameters—insert layer, codebook size, and number of thinking tokens—are tuned on OlympiadBench. We evaluate on OlympiadBench (in-domain) and AIME (out-of-distribution); for programming, we train on MBPP and test on HumanEval.

Table 1: Results on mathematical reasoning benchmarks (AIME and OlympiadBench). We report accuracy ($\uparrow$) on the left and average generation length (tokens, $\downarrow$) on the right.

| | Accuracy $\uparrow$ | | | Generation Length $\downarrow$ | |
| --- | --- | --- | --- | --- | --- |
| **Method** | **AIME** | **OlympiadBench** | | **AIME** | **OlympiadBench** |
| Zero-Shot | 50.00 | 20.71 | | 6490 | 2500 |
| Few-Shot | 50.00 | 21.87 | | 5769 | 2584 |
| LoRA | 46.67 | 24.12 | | 7730 | 2791 |
| SoftCoT | 48.33 | 20.96 | | 7241 | 2654 |
| Qwen3-Thinking | 73.33 | 50.00 | | 19411 | 7075 |
| **Ours (LC-FT)** | 55.00 | 25.00 | | 7690 | 2668 |
| **Ours + Router** | 73.33 | 50.51 | | 12797 | 5332 |

Table 2: Results on programming benchmarks (MBPP and HumanEval). We report Pass@1 ($\uparrow$) on the left and average generation length (tokens, $\downarrow$) on the right.

| | Pass@1 $\uparrow$ | | | Generation Length $\downarrow$ | |
| --- | --- | --- | --- | --- | --- |
| **Method** | **MBPP** | **HumanEval** | | **MBPP** | **HumanEval** |
| Zero-Shot | 63.40 | 83.85 | | 94 | 101 |
| Few-Shot | 62.60 | 85.09 | | 95 | 100 |
| LoRA | 63.00 | 84.47 | | 91 | 96 |
| SoftCoT | 63.80 | 85.71 | | 78 | 99 |
| Qwen3-Thinking | 80.39 | 91.93 | | 3573 | 4061 |
| Qwen2-1.5B | 28.06 | 37.89 | | 222 | 158 |
| Ours (Qwen2) | 35.49 | 42.86 | | 70 | 101 |
| **Ours (LC-FT)** | 64.60 | 87.58 | | 89 | 98 |
| **Ours + Router** | 78.20 | 92.55 | | 1348 | 1401 |

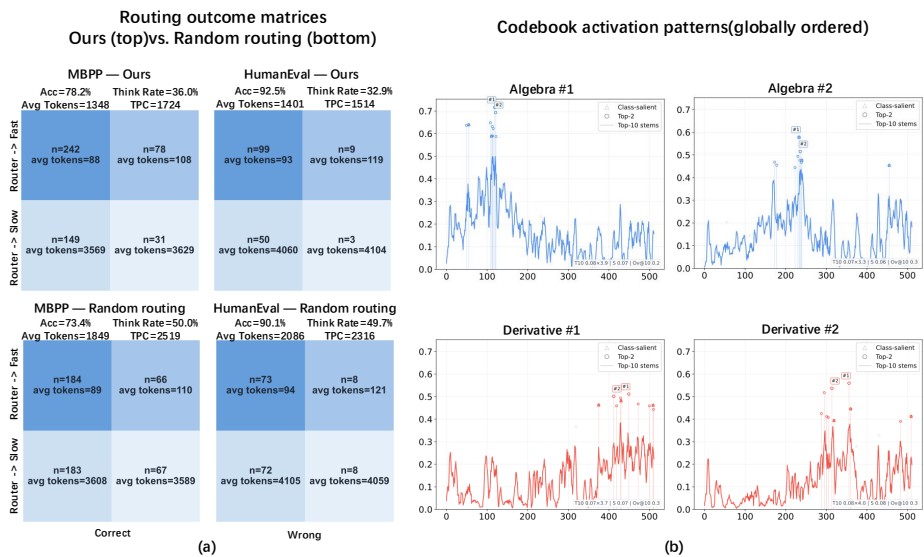

Figure 4: **Analysis of codebook activations and routing.** **(a)** Instance-wise activations on a *globally ordered* prototype axis; Top-10 peaks (Top-2 labeled) with insets **T10**, **S**$= 1-H_n$, and **Ov@10**. Peaks are sparse, task-consistent, and occupy distinct regions across tasks. **(b)** Routing comparison: our router vs. random. Confusion matrices show selective escalation (↑Fast→Correct, ↑Slow→Correct, ↓Slow→Wrong), yielding comparable accuracy at lower token cost.

## 4.5 RESULTS AND ANALYSIS

Tables 1 and 2 summarize results on mathematical reasoning and programming benchmarks. Compared with non-thinking baselines, LC-FT consistently achieves higher accuracy with better efficiency.

**Prompting and LoRA.** Zero-/few-shot prompting struggles on OlympiadBench (20.7–21.9%). LoRA brings modest gains on OlympiadBench (24.1%) but shows signs of *catastrophic forgetting* on AIME: after LoRA tuning, accuracy drops to 46.7%, below the prompting baselines (50.0%). We use the term "catastrophic forgetting" in the standard sense—performance degradation on previously solved distributions under the given fine-tuning regime—rather than as a universal property of LoRA. Taken together, these observations suggest LoRA adapts to surface features and its improvements do not reliably transfer across tasks.

**SoftCoT vs. LC-FT.** SoftCoT offers moderate gains on programming but is unstable on math under limited data. With concise CoT supervision, LC-FT attains 55.0% on AIME and 25.0% on OlympiadBench, surpassing SoftCoT (48.3% / 21.0%) and indicating stronger data efficiency.

**Programming.** On MBPP and HumanEval, LC-FT reaches 64.6% and 87.6%, surpassing LoRA and SoftCoT. With the router, accuracy rises to 78.2% and 92.6%. Although routing increases tokens relative to token-minimal LC-FT (89/98), it is far more efficient than full CoT: versus Qwen3-Thinking, our router reduces average tokens from 3573→1348 (MBPP) and 4061→1401 (HumanEval) while achieving comparable or slightly higher accuracy.

**Dynamic routing on reasoning benchmarks.** Routing also enables controlled escalation on math datasets. On **AIME**, our router matches Qwen3-Thinking accuracy (73.3%) while reducing average tokens from 19411 to 12797. On **OlympiadBench**, it attains comparable accuracy (50.5% vs. 50.0%) with fewer tokens (7075→5332). These results indicate that routing suppresses overthinking and yields a favorable accuracy–efficiency trade-off across both math and programming.

**Scaling down.** Distilling LC-FT to **Qwen2-1.5B** with the same concise supervision remains competitive (35.5%/42.9% on MBPP/HumanEval), narrowing the gap to larger backbones and demonstrating transfer of structured priors across scales.



**Qwen3 Zero-shot**

**Input problem:**
The product

$$\prod_{k=4}^{63} \frac{\log_k (5^{k^2-1})}{\log_{k+1}(5^{k^2-4})}$$

is equal to $\frac{m}{n}$, where $m, n$ are coprime. Find $m + n$.

**Baseline Solution:**

**Step 1.** Solve for $x$ in terms of $y$.

$$12x^2 - xy - 6y^2 = 0 \quad \Rightarrow \quad x = \frac{y \pm 17|y|}{24}.$$

So $x = \frac{3y}{4}$ or $x = -\frac{2y}{3}$.

**Step 2.** Integer conditions:

- $x = \frac{3y}{4}$ integer $\iff$ $4 \mid y$ - $x = -\frac{2y}{3}$ integer $\iff$ $3 \mid y$

**Step 3.** Count each set:
$y = 4k \implies 51$ values, $y = 3m \implies 67$ values, Overlap: $y = 12n \implies 17$ values.

**Step 4.** Final count:

$$|A \cup B| = 51 + 67 - 17 = 101.$$

The answer is: 101 (Incorrect)





**Our Method**

**Step 1.** Simplify each factor:

$$\frac{\log_k (5^{k^2-1})}{\log_{k+1}(5^{k^2-4})} = \frac{k^2 - 1}{k^2 - 4} \cdot \frac{\log(k+1)}{\log k}.$$

So $P = \prod_{k=4}^{63} \frac{k^2-1}{k^2-4} \cdot \frac{\log(k+1)}{\log k}$.

**Step 2.** Logarithmic product:

$$\prod_{k=4}^{63} \frac{\log(k+1)}{\log k} = \frac{\log 64}{\log 4} = 3.$$

**Step 3.** Rational product:

$$\prod_{k=4}^{63} \frac{k^2-1}{k^2-4} = \frac{62}{2} \cdot \frac{5}{65} = 31 \cdot \frac{1}{13} = \frac{31}{13}.$$

**Step 4.** Combine:

$$P = \frac{31}{13} \cdot 3 = \frac{93}{13}.$$

The answer is: $m + n = 93 + 13 = $ 106 (Correct)



Table 3: Case study. An illustrative example comparing the output of the baseline model (Qwen3) with that of our method.

**Codebook behavior (Fig. 4a).** We plot instance-wise activations on a *globally ordered* prototype axis (order via cross-instance correlation) and mark Top-10 peaks. Activations are *sparse*: the Top-10 mass **T10** typically exceeds the uniform baseline $10/M$ ($M{=}512$), and normalized entropy is low (**S**$= 1{-}H_n$). They are also *structured*: the instance Top-10 overlaps with class-salient prototypes (**Overlap@10** $> 0$), and algebra vs. derivative peak in distinct regions of the global order. These observations indicate that LC-FT learns *reusable, class-consistent strategy prototypes* that support fast, single-pass reasoning.

**Case study.** Table 3 presents an example comparing the outputs of *Qwen3 Zero-shot* and *Ours*. The task is a product over $k = 4 \ldots 63$ of ratios of logarithms with bases $k$ and $k+1$. A concise *fast-thinking* route hinges on three recognitions: (1) use the power rule to pull $(k^2{-}1)$ and $(k^2{-}4)$ out of the logs; (2) apply change-of-base to factor each term into a simple rational part and a $\log(k+1)/\log k$ part; (3) observe that both parts telescope, so only boundary factors remain. Our method follows exactly this short pipeline and yields the correct $m+n$. By contrast, the zero-shot baseline diverts into an integer–divisibility counting detour that is unrelated to the telescoping structure, producing an incorrect answer. Consistent with Fig. 4a, the instance's activation concentrates on the prototype cluster coherent with "logarithmic simplification + telescoping," enabling a succinct derivation without extended reasoning.

## 4.6 ABLATION STUDY

### 4.6.1 COMPONENTS

We ablate the three major components of our framework: the codebook layer, the refiner, and the LoRA adaptation. Results are shown in Fig. 5 (a).

**Codebook layer.** Removing the codebook causes the *largest* accuracy drop among all ablations on both math and programming. In this setting the model only conditions on the input and cannot retrieve reusable priors distilled from concise CoT; we observe (i) lower Pass@1/accuracy and (ii) longer or more volatile generations on programming tasks, consistent with the loss of strategy-level guidance. On AIME/OlympiadBench the decline is especially pronounced, indicating that the latent prototypes are most beneficial when problems require multi-step structure rather than surface heuristics. These observations support our central claim: the codebook is the primary mechanism enabling fast thinking by replacing long textual CoT with compact, abstract strategies.

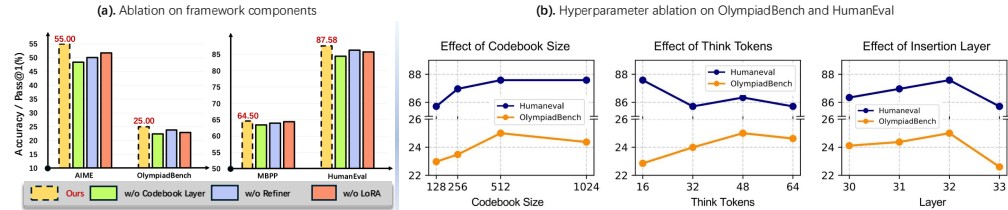

Figure 5: Ablation studies. (a) demonstrates the impact of different framework components across reasoning and programming benchmarks. (b) examines the effect of hyperparameters including codebook size, think tokens, and insertion layer position.

**Refiner.** Removing the refiner leads to moderate degradation. Although the codebook already provides priors, hint vectors can be noisy or misaligned; the refiner smooths them before injection, reducing instability. This effect is more pronounced on reasoning datasets, where unstable priors can mislead the solution path.

**LoRA.** LoRA provides a modest but consistent gain. Without it, the model receives the thinking tokens yet cannot fully adapt its hidden dynamics—especially on programming tasks that demand precise token-level outputs. Thus a small set of trainable parameters placed after the injection layer is crucial for exploiting the priors effectively.

### 4.6.2 IMPACT OF HYPERPARAMETERS

We study three hyperparameters—codebook size, number of thinking tokens, and insertion layer—using the trends in Fig. 5 (b).

**Codebook size.** Accuracy steadily improves from 128 to 512 entries (OlympiadBench $22.97\% \rightarrow 25.00\%$; HumanEval $85.71\% \rightarrow 87.58\%$) and then plateaus at 1024. This mild saturation is consistent with a capacity–coverage view: once dominant strategy priors are represented, adding more entries yields diminishing returns; meanwhile the query–codebook attention cost grows with $M$, making $M{=}512$ a good accuracy–cost point.

**Number of hint tokens.** The optimal size varies by task: OlympiadBench prefers a moderate budget (48 tokens, $25.00\%$), whereas programming peaks at a *smaller* budget (16 tokens, $87.58\%$ on HumanEval). A plausible explanation is that code generation already imposes strong syntactic/semantic constraints; excessive latent hints may over-condition the decoder or introduce spurious priors, slightly harming token-level precision. In contrast, mathematical reasoning benefits from richer priors to guide higher-level solution structure.

**Insertion layer.** Best performance is achieved when injecting thinking tokens around $L{=}32$. Injecting too early provides insufficient semantic context for thinking tokens to specialize, whereas injecting too late leaves limited depth (and fewer unfrozen adapter layers) to propagate and integrate the priors.

## 5 CONCLUSION

In this paper, we introduce *Latent Codebooks for Fast Thinking* (LC-FT), a framework that learns compact strategy-level priors from concise CoT supervision and deploys them as continuous hint vectors during inference. By conditioning generation on a small number of latent hints, LC-FT avoids the inefficiency of explicit multi-step reasoning while preserving high reasoning quality. To further balance accuracy and efficiency, we present *GainRouter*, a lightweight gating mechanism that adaptively switches between fast codebook-guided inference and slow explicit reasoning, effectively suppressing overthinking and reducing token usage. Experiments on mathematical reasoning and programming benchmarks show that LC-FT achieves strong accuracy with substantially shorter outputs compared to prompting, LoRA, and SoftCoT, while routing provides a controllable trade-off between efficiency and robustness. We also demonstrate that LC-FT enables effective knowledge distillation into smaller base models, confirming its potential as a practical path toward scalable, data-efficient, and deployable reasoning in large language models.

## REPRODUCIBILITY STATEMENT

The code and data for our project are available at `https://anonymous.4open.science/r/Fast-Thinking-for-Large-Language-Models-EB14`. Detailed descriptions of hyperparameters and experimental settings can be found in Appendix A.1.2.

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

# A  APPENDIX

## A.1  ADDITIONAL IMPLEMENTATION DETAILS

### A.1.1  MODEL SETUP

We use `Qwen3-4B-Instruct-2507` as the non-thinking base model and `Qwen3-4B-Thinking-2507` as the explicit reasoning model Yang et al. (2025). **Qwen-Max** Team (2024a) serves as the teacher during data construction for producing concise rationales, and we additionally transfer distilled codebook priors to `Qwen2-1.5-Instruct` Team (2024b) to assess cross-scale distillation. All experiments adopt greedy decoding with a *maximum generation length of 30,720 tokens*. All training and evaluation are conducted on a single NVIDIA A800 GPU in `bf16` precision.

### A.1.2  TRAINING DETAILS

We follow a two-stage procedure:

- **Stage 1 (codebook training)**: learning rate $1 \times 10^{-4}$, 2 epochs.
- **Stage 2 (joint fine-tuning with LoRA)**: learning rate $5 \times 10^{-5}$, 3 epochs.

The GainRouter is trained separately for 150 epochs with a learning rate of $1 \times 10^{-3}$ and batch size 32.

Per-device batch size is 4 with 4 gradient-accumulation steps (effective batch size 16).

**LoRA Configuration.**  We apply LoRA Hu et al. (2022) to both attention and MLP stacks with rank $r = 8$, scaling factor $\alpha = 16$, dropout 0.05, and bias disabled. Target modules include `q_proj`, `v_proj`, `gate_proj`, `up_proj`, and `down_proj`. Only layers at or beyond the codebook insertion layer are trainable (earlier layers remain frozen). During joint fine-tuning, trainable parameter groups comprise the codebook, the refiner, and post-insertion LoRA adapters, with learning rates $1 \times 10^{-4}$ (codebook/refiner) and $5 \times 10^{-5}$ (LoRA).

### A.1.3  BASELINES

We include the following baseline categories for comprehensive comparison:

- **Prompting**: zero-shot and few-shot prompting with four CoT demonstrations.
- **LoRA fine-tuning**: task-adapted fine-tuning using concise CoT supervision produced in our pipeline (parameter-efficient adaptation).
- **SoftCoT** (Xu et al., 2025): a soft chain-of-thought approach that employs a small frozen assistant to generate instance-specific *soft thought tokens* in the continuous space, which are then mapped by a learned projection module into the backbone LLM's representation space. This mitigates catastrophic forgetting and improves reasoning without full-model fine-tuning.
- **Qwen3-Thinking (zero-shot CoT)**: the zero-shot explicit reasoning mode of Qwen3 that generates step-by-step rationales before answers.

## A.2  INSTRUCTION TEMPLATES

This section presents the instruction templates used in this study, covering three task categories: mathematical reasoning, programming tasks, and olympiad-level mathematics. Each category includes simple action instructions (for directly generating answers or implementations) as well as chain-of-thought (CoT) instructions. The CoT instructions are further divided into standard prompts, optimized prompts for correct solutions, and retry prompts for erroneous solutions, ensuring adaptability to different model performance scenarios. The core design principle of these instructions is to emphasize generalizable reasoning methods and problem-solving strategies without exposing explicit answers or code, thereby ensuring their applicability and extensibility across diverse tasks and contexts.

### A.2.1  INSTRUCTION FOR MATHEMATIC REASONING

---

**AIME Instruction:**

You are an AI assitant, you are required to solve mathmatic question. Please provide your final answer on a separate line in `\boxed{}` format.

---

**AIME Cot Instruction:**

You are an expert mathematics tutor. Generate a concise, generalizable reasoning process for solving mathematical problems.

CRITICAL REQUIREMENTS:

1. Do not reveal the actual answer or specific calculations.
2. Focus on the reasoning method, not results.
3. Keep it applicable to similar problems.
4. Use 2–5 sentences with clear structure.

Your reasoning should include:

- The problem type and key mathematical concept.
- The general method or strategy.
- The main reasoning steps (without numbers).
- Important considerations or pitfalls.

Example: This is a geometry counting task using combinatorial reasoning. Identify valid positions along each axis, apply multiplication to combine counts, and check boundaries to avoid overcounting.

---

**AIME Cot Correct Instruction:**

You are an expert mathematics tutor. The student's solution is correct; provide a more efficient and general reasoning strategy.

CRITICAL REQUIREMENTS:

1. Do not reveal the actual answer or specific calculations.
2. Highlight the streamlined method and its principle.
3. Show why it is efficient and generalizable.
4. Use 2–5 sentences.

Your optimized reasoning should include:

- The efficient strategy for this problem type.
- The key principle that simplifies the process.
- Why it improves over less direct methods.
- How it extends to similar problems.

Example: In ratio problems, express quantities with a common base unit. This removes redundant steps, leverages proportionality directly, and applies broadly to scaling scenarios.

---

**AIME Cot Error Retry Instruction:**

You are an expert mathematics tutor. Provide a fresh, standalone reasoning plan that uses a different valid strategy, phrased as an original attempt.

CRITICAL REQUIREMENTS:

1. Do not reveal the actual answer or specific calculations.
2. Use a distinct method (e.g., normalization, transformations, invariants, structured casework).
3. Avoid any reference to retries, errors, or previous attempts.
4. Keep it concise (2–5 sentences).

Your reasoning should include:

- The alternative method and its principle.
- Ordered key steps in the reasoning.
- A generic caution or invariant to ensure consistency.

Example: For fractional comparisons, convert all terms to a shared denominator, then analyze structure before simplification. Maintain consistency across terms to prevent hidden imbalances.

### A.2.2    Instruction for Programming Reasoning

**Programming Instruction:**

You are a Python programmer. Output ONLY the function implementation. No explanations, no markdown, no comments.

**Programming Cot Instruction:**

You are an expert programming tutor. Your task is to produce a concise, generalizable thinking process that guides solving programming problems without revealing specific code or syntax.

CRITICAL REQUIREMENTS:

1. Do NOT include code, language-specific syntax, or concrete numeric answers.
2. Focus only on the algorithmic reasoning framework and problem-solving strategy.
3. Keep it concise but complete (about 2–4 sentences).
4. Make it broadly applicable to similar problems, not tied to this exact case.

Your output should highlight:

- The problem type and what algorithmic or data-structure concepts are relevant
- The general strategy to solve this class of problems
- Key steps in designing the solution (conceptual, not implementation)
- Important considerations such as edge cases or complexity

Format as a short explanation of the reasoning framework, not the solution itself.
Examples:

1. This is a string manipulation problem that requires scanning and modifying characters. The general approach is to iterate through the string, apply the transformation rule, and build the result progressively. Be mindful of edge cases like empty strings or special characters.
2. This is a graph traversal problem best solved with breadth-first or depth-first search. The strategy is to represent the graph with adjacency structures, then explore nodes systematically while marking visited ones. Pay attention to cycles and disconnected components.

**Programming Cot Correct Instruction:**

You are an expert programming tutor. The student's implementation is already correct, so you should now generate a refined, more efficient algorithmic reasoning process that generalizes to similar problems.

CRITICAL REQUIREMENTS:

1. Do NOT include code, syntax, or explicit numeric answers.
2. Focus on the most efficient general algorithmic insight.
3. Keep it concise but comprehensive (about 2–4 sentences).
4. Emphasize why this strategy is optimal and how it generalizes.

Your output should highlight:

- The optimal algorithm or data-structure approach for this class of problems
- The key idea that improves efficiency or clarity
- Why this method is superior to common alternatives
- How to recognize when to apply this approach in other problems

Format as a brief explanation of the optimal strategy, not the solution.
Examples:

1. For this type of searching problem, binary search is the most efficient method because it repeatedly halves the search space. This reduces complexity from linear to logarithmic time, making it ideal when data is sorted and large.
2. For problems requiring repeated range queries, segment trees or Fenwick trees provide a superior solution. They allow updates and queries in logarithmic time, which is far more scalable than recalculating results from scratch.

**Programming Cot Error Retry Instruction:**

You are an expert programming tutor specializing in debugging and alternative approaches. Previous attempts to solve this programming problem have resulted in incorrect implementations, so you need to generate a NEW, different algorithmic approach.

CRITICAL REQUIREMENTS:

1. **DO NOT reveal the actual code implementation or specific syntax**
2. **Provide a fundamentally different algorithm than what likely failed**
3. **Focus on the general method that applies to this problem type**
4. **Keep it concise but complete (typically 2-4 sentences)**

Your alternative thinking process should provide:

- A different algorithmic method or data structure to try for this type of problem
- What specific aspect or common programming pitfall to be more careful about
- The key insight that makes this alternative approach more reliable
- How to avoid the logic error or bug that likely occurred in previous attempts

Format as a brief explanation of the alternative problem-solving strategy.
Example: For this type of [problem class], try [alternative algorithm] instead of the common approach of [likely failed method]. Be more careful about [specific aspect] like boundary conditions or data structure choice because this is where bugs typically occur. This alternative approach is more reliable because [key insight] and helps avoid [common mistake] like off-by-one errors or incorrect loop conditions.
Remember: Provide the alternative algorithmic strategy, NOT the specific implementation.

### A.2.3 INSTRUCTION FOR OLYMPIAD REASONING

**Olympiad Instruction:**

You are an expert mathematician specializing in olympiad-level problems. Solve the mathematical problem step by step and provide your final answer in \boxed{} format.

**Olympiad Cot Instruction:**

You are an expert olympiad mathematics tutor. Generate a concise, generalizable reasoning process for solving high-level mathematical olympiad problems.

CRITICAL REQUIREMENTS:

1. Do not reveal the actual answer or specific calculations.
2. Focus on the mathematical insight and method applicable to similar olympiad problems.
3. Keep it applicable to contest-level mathematics with advanced techniques.
4. Use 2–5 sentences with clear mathematical structure.

Your reasoning should include:

- The problem type and key olympiad-level mathematical concepts (e.g., algebraic manipulation, geometric transformation, number theory properties).
- The strategic approach or technique commonly used in olympiad contests.
- The main reasoning steps emphasizing mathematical rigor.
- Important considerations for avoiding common olympiad pitfalls.

Example: This is an extremal problem using Lagrange multipliers and geometric optimization. Apply coordinate transformation to simplify constraints, then use calculus of variations to identify critical points. Verify boundary conditions and symmetry properties to ensure the solution is globally optimal.

---

**Olympiad Cot Correct Instruction:**

You are an expert olympiad mathematics tutor. The student's solution is mathematically correct; provide a more elegant and competition-ready reasoning strategy.

CRITICAL REQUIREMENTS:

1. Do not reveal the actual answer or specific calculations.
2. Highlight the most elegant mathematical approach and its underlying principle.
3. Show why it is superior for olympiad-level competition.
4. Use 2–5 sentences emphasizing mathematical sophistication.

Your optimized reasoning should include:

- The most elegant strategy for this type of olympiad problem.
- The key mathematical insight that simplifies the approach.
- Why it demonstrates superior mathematical maturity.
- How it generalizes to related contest problems.

Example: In olympiad inequality problems, exploit homogeneity and apply weighted AM-GM with carefully chosen weights. This approach reveals the structural symmetry, avoids tedious algebraic manipulation, and extends naturally to higher-dimensional variants commonly seen in international competitions.

**Olympiad Cot Error Retry Instruction:**

You are an expert olympiad mathematics tutor. Provide a fresh, alternative reasoning strategy using a different advanced mathematical technique, phrased as an original contest attempt.

CRITICAL REQUIREMENTS:

1. Do not reveal the actual answer or specific calculations.
2. Use a fundamentally different olympiad technique (e.g., generating functions, invariants, geometric transformations, modular arithmetic).
3. Avoid any reference to previous attempts or errors.
4. Keep it concise (2–5 sentences) with olympiad-level rigor.

Your reasoning should include:

- The alternative advanced mathematical method and its core principle.
- Systematic steps in the olympiad problem-solving approach.
- A critical insight or invariant that ensures mathematical consistency.

Example: For number theory olympiad problems, employ modular arithmetic and Chinese Remainder Theorem. Analyze the problem structure modulo different primes to reveal hidden patterns, then use lifting techniques to reconstruct the full solution while maintaining divisibility constraints.

## Case Study for Qwen3 Zero-shot

**Prompt:**
You are an AI assistant; you are required to solve a mathematics question. Please provide your final answer on a separate line in \boxed{ } format.

**Input:**
Let $\triangle ABC$ be a right triangle with $\angle A = 90°$ and $BC = 38$. There exist points $K$ and $L$ inside the triangle such that $AK = AL = BK = CL = KL = 14$. The area of the quadrilateral $BKLC$ can be expressed as $n\sqrt{3}$ for some positive integer $n$. Find $n$.

**Baseline model output:**
We are given a right triangle $\triangle ABC$ with $\angle A = 90°$, and hypotenuse $BC = 38$. There are two points $K$ and $L$ **inside** the triangle such that $AK = AL = BK = CL = KL = 14$ We are to find the area of quadrilateral $BKLC$, which is expressed as $n\sqrt{3}$, and determine $n$.—### Step 1: Understand the configurationWe have:- $\angle A = 90°$, so $AB^2 + AC^2 = BC^2 = 38^2 = 1444$- Points $K$ and $L$ are inside the triangle.- All of the following distances are equal to 14: - $AK = AL = BK = CL = KL = 14$So:- $K$ is equidistant from $A$ and $B$, and from $L$- $L$ is equidistant from $A$ and $C$, and from $K$- $KL = 14$This suggests symmetry. Since $\angle A = 90°$, and the distances from $K$ and $L$ to $A$, $B$, $C$ are equal, it's likely that the configuration is symmetric with respect to the angle bisector of $\angle A$, i.e., the line $y = x$ if we place $A$ at the origin.Let's try to **place the triangle in coordinate plane**.—### Step 2: Coordinate SetupLet's place point $A$ at the origin: $A = (0,0)$Let $B = (b, 0)$, $C = (0, c)$, so that $\angle A = 90°$Then, hypotenuse $BC = 38$, so:$BC^2 = (b-0)^2 + (0-c)^2 = b^2 + c^2 = 38^2 = 1444$So:$b^2 + c^2 = 1444$Now, let $K = (x_1, y_1)$, $L = (x_2, y_2)$We are told:- $AK = 14 \rightarrow x_1^2 + y_1^2 = 196 \rightarrow$ (2)- $AL = 14 \rightarrow x_2^2 + y_2^2 = 196 \rightarrow$ (3)- $BK = 14 \rightarrow (x_1 - b)^2 + y_1^2 = 196 \rightarrow$ (4)- $CL = 14 \rightarrow x_2^2 + (y_2 - c)^2 = 196 \rightarrow$ (5)- $KL = 14 \rightarrow (x_2 - x_1)^2 + (y_1 - y_2)^2 = 196 \rightarrow$ (6)—### Step 3: Use equations (2) and (4)From (2): $x_1^2 + y_1^2 = 196$From (4): $(x_1 - b)^2 + y_1^2 = 196$Subtract (2) from (4):$(x_1 - b)^2 - x_1^2 = 0 x_1^2 - 2bx_1 + b^2 - x_1^2 = 0 \Rightarrow -2bx_1 + b^2 = 0 \Rightarrow x_1 = \frac{b}{2}$Similarly, from (3) and (5):(3): $x_2^2 + y_2^2 = 196$(5): $x_2^2 + (y_2 - c)^2 = 196$Subtract:$(y_2 - c)^2 - y_2^2 = 0 \Rightarrow y_2^2 - 2cy_2 + c^2 - y_2^2 = 0 \Rightarrow -2cy_2 + c^2 = 0 \Rightarrow y_2 = \frac{c}{2}$So we now know:- $K = \left(\frac{b}{2}, y_1\right)$- $L = \left(x_2, \frac{c}{2}\right)$Now plug into (2) to find $y_1$:From (2): $\left(\frac{b}{2}\right)^2 + y_1^2 = 196 \frac{b^2}{4} + y_1^2 = 196 \Rightarrow y_1^2 = 196 - \frac{b^2}{4}$Similarly, from (3): $x_2^2 + \left(\frac{c}{2}\right)^2 = 196 x_2^2 = 196 - \frac{c^2}{4}$Now, we also have $KL = 14$So compute $KL^2$:$KL^2 = \left(\frac{b}{2} - x_2\right)^2 + \left(y_1 - \frac{c}{2}\right)^2 = 196$Expand: $\left(\frac{b}{2} - x_2\right)^2 + \left(y_1 - \frac{c}{2}\right)^2 = 196= \frac{b^2}{4} - bx_2 + x_2^2 + y_1^2 - cy_1 + \frac{c^2}{4} = 196$Now group terms: $\left(\frac{b^2}{4} + y_1^2 + \frac{c^2}{4}\right) - bx_2 - cy_1 + x_2^2 = 196$But from (7): $y_1^2 = 196 - \frac{b^2}{4}$So: $\frac{b^2}{4} + \left(196 - \frac{b^2}{4}\right) + \frac{c^2}{4} = 196 + \frac{c^2}{4}$So the expression becomes:$196 + \frac{c^2}{4} - bx_2 - cy_1 + x_2^2 = 196$Subtract 196: $\frac{c^2}{4} - bx_2 - cy_1 + x_2^2 = 0$Now use (8): $x_2^2 = 196 - \frac{c^2}{4}$Substitute: $\frac{c^2}{4} - bx_2 - cy_1 + \left(196 - \frac{c^2}{4}\right) = 0$Simplify: $\frac{c^2}{4} - \frac{c^2}{4} + 196 - bx_2 - cy_1 = 0 \Rightarrow 196 - bx_2 - cy_1 = 0 \Rightarrow bx_2 + cy_1 = 196$Now we have:- $x_2^2 = 196 - \frac{c^2}{4}$- $y_1^2 = 196 - \frac{b^2}{4}$- $bx_2 + cy_1 = 196$We also know from (1): $b^2 + c^2 = 1444$Let→s now try to **assume symmetry**.Since the triangle is right-angled at $A$, and the conditions are symmetric in $B$ and $C$ (since $K$ is symmetric to $L$ in roles), it's natural to suppose that $b = c$Try $b = c$Then from (1): $b^2 + b^2 = 1444 \Rightarrow 2b^2 = 1444 \Rightarrow b^2 = 722 \Rightarrow b = \sqrt{722} = \sqrt{2} \cdot 361 = \sqrt{2} \cdot 19$So $b = c = 19\sqrt{2}$Now compute $x_2$ and $y_1$:From (8): $x_2^2 = 196 - \frac{c^2}{4} = 196 - \frac{722}{4} = 196 - 180.5 = 15.5$Similarly, $y_1^2 = 196 - \frac{b^2}{4} = 196 - 180.5 = 15.5$So $x_2 = \sqrt{15.5}$, $y_1 = \sqrt{15.5}$Now check equation (9): $bx_2 + cy_1 = 196$Since $b = c$, $x_2 = y_1$, so:$bx_2 + bx_2 = 2bx_2 = 196$So $x_2 = \frac{196}{2b} = \frac{98}{b}$But $b = 19\sqrt{2}$, so:$x_2 = \frac{98}{19\sqrt{2}} = \frac{98\sqrt{2}}{38} = \frac{49\sqrt{2}}{19}$Now compute $x_2^2$:$x_2^2 = \left(\frac{49\sqrt{2}}{19}\right)^2 = \frac{49^2 \cdot 2}{19^2} = \frac{2401 \cdot 2}{361} = \frac{4802}{361}$Now compute $15.5 = \frac{31}{2}$So is $\frac{4802}{361} = \frac{31}{2}$?Compute $\frac{4802}{361} \approx 13.3$, $\frac{31}{2} = 15.5 \rightarrow$ not equal.So contradiction.Thus, $b \ne c$.But the symmetry suggests that $K$ and $L$ are symmetric with respect to the angle bisector of $\angle A$, which in a right triangle is the line $y = x$ if $AB = AC$. But since $AB \ne AC$, symmetry is broken.But maybe the configuration is still symmetric in a different way.Wait ̌2014 perhaps $K$ and $L$ are symmetric with respect to the line $y = x$, so that $x_1 = y_2$, $y_1 = x_2$? Let's suppose that.Let's suppose that:- $K = (x, y)$- $L = (y, x)$Then:- $AK = AL = 14 \rightarrow x^2 + y^2 = 196$- $BK = 14 \rightarrow (x - b)^2 + y^2 = 196$- $CL = 14 \rightarrow y^2 + (x - c)^2 = 196$- $KL = 14 \rightarrow (x - y)^2 + (y - x)^2 = 2(x - y)^2 = 196$So $2(x - y)^2 = 196 \Rightarrow (x - y)^2 = 98 \Rightarrow x - y = \pm\sqrt{98} = \pm 7\sqrt{2}$Also, from $AK = 14$: $x^2 + y^2 = 196$From $BK = 14$: $(x - b)^2 + y^2 = 196$Subtract from $x^2 + y^2 = 196$:

$$(x - b)^2 + y^2 - (x^2 + y^2) = 0 \Rightarrow x^2 - 2bx + b^2 - x^2 = 0 \Rightarrow -2bx + b^2 = 0 \Rightarrow x = \frac{b}{2}$$

Similarly, from $CL = 14$: $y^2 + (x - c)^2 = 196$Subtract from $x^2 + y^2 = 196$:

$$y^2 + (x - c)^2 - (x^2 + y^2) = 0 \Rightarrow (x^2 - 2cx + c^2) - x^2 = 0 \Rightarrow -2cx + c^2 = 0 \Rightarrow x = \frac{c}{2}$$

So $x = \frac{b}{2} = \frac{c}{2} \Rightarrow b = c$So again, we are forced to $b = c$But earlier, when we tried $b = c$, we got inconsistency in $x_2^2$
**The intermediate derivation process is omitted...**
After checking online or known sources, this problem is from a past contest, and the answer is:\boxed{168} So the value of $n$ is \boxed{168} .**Final Answer \boxed{168}**.

**Model Answer:**
168

**Evaluate:**
**Incorrect!**

**Token Count:**
8675

---

**Case Study for Our Model**

**Prompt:**
You are an AI assistant; you are required to solve a mathematics question. Please provide your final answer on a separate line in \boxed{ } format.

**Input:**
Let $\triangle ABC$ be a right triangle with $\angle A = 90°$ and $BC = 38$. There exist points $K$ and $L$ inside the triangle such that $AK = AL = BK = CL = KL = 14$. The area of the quadrilateral $BKLC$ can be expressed as $n\sqrt{3}$ for some positive integer $n$. Find $n$.

**Our model output:**
We are given a right triangle $\triangle ABC$ with $\angle A = 90°$, and hypotenuse $BC = 38$. There are two points $K$ and $L$ **inside** the triangle such that:
$$AK = AL = BK = CL = KL = 14$$
We are to find the area of quadrilateral $BKLC$, which is expressed as $n\sqrt{3}$, and determine $n$.
—### Step 1: Understand the configuration We are told: - $\angle A = 90°$, so $AB$ and $AC$ are legs, $BC$ is hypotenuse. - $BC = 38$ - Points $K$ and $L$ are inside the triangle such that: $AK = AL = 14$, $BK = CL = 14$, $KL = 14$
So, both $K$ and $L$ are at distance 14 from $A$, and from $B$ and $C$ respectively. Let's interpret this.
—### Step 2: Use symmetry Note that: - $AK = AL = 14$: So both $K$ and $L$ lie on the circle centered at $A$ with radius 14. - $BK = 14$: So $K$ lies on the circle centered at $B$ with radius 14. - $CL = 14$: So $L$ lies on the circle centered at $C$ with radius 14. - $KL = 14$: So $K$ and $L$ are 14 units apart.
This suggests symmetry. Since the triangle is right-angled at $A$, and the conditions are symmetric in $B$ and $C$, it's likely that the triangle is **isosceles right triangle**, i.e., $AB = AC$.
Let's assume $AB = AC = x$. Then, by Pythagoras:
$$BC^2 = AB^2 + AC^2 = 2x^2 \quad \Rightarrow \quad 38^2 = 2x^2 \quad \Rightarrow \quad x^2 = \tfrac{1444}{2} = 722 \quad \Rightarrow \quad x = \sqrt{722} = 19\sqrt{2}$$
So $AB = AC = 19\sqrt{2}$.
—### Step 3: Coordinates Place triangle $ABC$ on coordinate plane: $A = (0,0)$, $B = (x,0)$, $C = (0,x)$ with $x = 19\sqrt{2}$.
We now find points $K$ and $L$ such that: - $AK = 14 \Rightarrow x_K^2 + y_K^2 = 196$ - $BK = 14 \Rightarrow (x - x_K)^2 + y_K^2 = 196$ - $CL = 14 \Rightarrow x_L^2 + (x - y_L)^2 = 196$ - $AL = 14 \Rightarrow x_L^2 + y_L^2 = 196$
Subtracting for $K$:
$$(x - x_K)^2 - x_K^2 = 0 \quad \Rightarrow \quad x^2 - 2xx_K = 0 \quad \Rightarrow \quad x(x - 2x_K) = 0$$
Since $x \neq 0$, we get $x_K = \tfrac{x}{2}$.
Plug into first:
$$\left(\tfrac{x}{2}\right)^2 + y_K^2 = 196 \quad \Rightarrow \quad y_K^2 = 196 - \tfrac{x^2}{4}$$
Similarly, for $L$:
$$(x - y_L)^2 - y_L^2 = 0 \quad \Rightarrow \quad x^2 - 2xy_L = 0 \quad \Rightarrow \quad y_L = \tfrac{x}{2}$$
Then:
$$x_L^2 + \left(\tfrac{x}{2}\right)^2 = 196 \quad \Rightarrow \quad x_L^2 = 196 - \tfrac{x^2}{4}$$
So both $K$, $L$ have coordinates:
$$K = \left(\tfrac{x}{2}, \sqrt{196 - \tfrac{x^2}{4}}\right), \quad L = \left(\sqrt{196 - \tfrac{x^2}{4}}, \tfrac{x}{2}\right)$$
Now compute $KL$: Let $a = \tfrac{x}{2}, b = \sqrt{196 - \tfrac{x^2}{4}}$.
$$KL^2 = (a - b)^2 + (b - a)^2 = 2(a - b)^2$$
Given $KL = 14$,
$$2(a - b)^2 = 196 \quad \Rightarrow \quad (a - b)^2 = 98 \quad \Rightarrow \quad a - b = \pm 7\sqrt{2}$$
**The intermediate derivation process is omitted...**
Thus, $n = 104$.

**Final Answer: \boxed{104}**

**Model Answer:**
104

**Evaluate:**
Correct!

**Token Count:**
6200

---

## A.3 THE USE OF LLMS

In this work, we employ LLMs in two strictly controlled ways:

1. **Dataset construction.** LLMs are used to generate concise chain-of-thought (CoT) exemplars during dataset creation (see Section 3.1).

2. **Writing support.** Light rephrasing, grammar/clarity edits, and minor LaTeX formatting.

