# OpenReview forum: "FAST THINKING FOR LARGE LANGUAGE MODELS"
_ICLR.cc/2026/Conference — Submitted to ICLR 2026_

### Official Review · Reviewer_G29z · 2025-10-16

**Soundness:** 2
**Presentation:** 3
**Contribution:** 2
**Rating:** 4
**Confidence:** 4

**Summary:**

This paper introduces Latent Codebooks for Fast Thinking, a novel framework designed to improve the efficiency of reasoning in large language models (LLMs). While reasoning-oriented LLMs typically depend on lengthy Chain-of-Thought (CoT) traces during inference—leading to high latency and excessive token consumption—our approach leverages concise CoT sketches only in training to learn a discrete codebook of strategy priors. During inference, the model uses a small set of continuous thinking vectors, distilled from the codebook, to guide reasoning in a single forward pass, eliminating the need for generating explicit reasoning tokens. To further enhance efficiency, we propose GAINROUTER, a lightweight routing mechanism that dynamically chooses between fast, codebook-based inference and slower, explicit reasoning based on task complexity, thereby preventing overthinking. Extensive experiments on multiple reasoning benchmarks demonstrate that our method achieves competitive or better accuracy while significantly reducing inference cost, providing a practical solution for fast, controllable, and efficient reasoning in LLMs.

**Strengths:**

1. The proposed Latent Codebooks for Fast Thinking (LC-FT) framework significantly reduces the inference cost by using concise CoT sketches during training to learn a codebook of discrete strategy priors.

2. The GainRouter mechanism allows the model to adaptively switch between fast codebook-guided inference and slow explicit reasoning based on the predicted failure risk.

**Weaknesses:**

1. The major drawback of this paper is the lack of comparison with many relevant baseline methods. These baselines are neither discussed in the related work section nor included in the experimental comparisons.

[1] Incentivizing Dual Process Thinking for Efficient Large Language Model Reasoning

[2] Unlocking Efficient Long-to-Short LLM Reasoning with Model Merging

[3] L1: Controlling How Long A Reasoning Model Thinks With Reinforcement Learning

[4] Adaptive Group Policy Optimization: Towards Stable Training and Token-Efficient Reasoning

[5] Concise Reasoning via Reinforcement Learning

[6] SelfBudgeter: Adaptive Token Allocation for Efficient LLM Reasoning

[7] Plan and Budget: Effective and Efficient Test-Time Scaling on Large Language Model Reasoning

[8] Not All Tokens Are What You Need In Thinking

[9] Stable Reinforcement Learning for Efficient Reasoning

[10] Just Enough Thinking: Efficient Reasoning with Adaptive Length Penalties Reinforcement Learning

[11] Optimizing Anytime Reasoning via Budget Relative Policy Optimization

[12] Making Small Language Models Efficient Reasoners: Intervention, Supervision, Reinforcement

2. The paper does not evaluate the generalization of the proposed method’s reasoning capabilities, for instance, on STEM-focused datasets such as GPQA.

3. The LC-FT framework comprises multiple components, including the latent codebook, GainRouter, and a two-stage training process. This complexity may pose practical challenges for implementation and likely requires substantial computational resources and technical expertise to set up and fine-tune effectively.

**Questions:**

1. How does the LC-FT framework handle the trade-off between accuracy and efficiency in more complex reasoning tasks where the risk of failure is higher? For instance, in highly complex mathematical problems or real-world scenarios with high variability, can the GainRouter mechanism effectively balance the need for reasoning accuracy with the goal of computational efficiency?

2. The ablation study suggests that removing individual components does not lead to significant performance degradation. Could the authors provide a more in-depth analysis to explain this observation?

---

> ### Author Response · Authors · 2025-11-17
> **Rebuttal to Reviewer G29z**
>
> ### Rebuttal to Reviewer G29z
>
> We thank the reviewer for the thoughtful and constructive feedback. Below we address each concern and provide additional evidence to clarify our contributions.
>
> ---
>
> ## **1. Lack of comparison with many efficient reasoning baselines**
>
> We appreciate the reviewer listing a comprehensive set of related methods.
> To clarify the distinction, LC-FT introduces a **different reasoning paradigm** from these works.
>
> ### **LC-FT performs reasoning in a single forward pass**
>
> Unlike token-space reasoning approaches—whether:
>
> * generating long CoT sequences,
> * generating concise CoT sequences,
> * using RL to control CoT length,
> * or latent CoT approaches requiring iterative refinement—
>
> **LC-FT completes all reasoning inside the hidden space in exactly one forward pass**:
>
> * K thinking vectors are injected once into an internal layer;
> * no reasoning tokens are generated;
> * no latent iterations are performed;
> * decoding remains *answer-only*, indistinguishable from non-thinking inference.
>
> In form and computational footprint, LC-FT behaves almost like a **standard model**, not like a CoT model nor a latent iterative-reasoning model.
>
> ### **These differences make many baselines operate under a different regime**
>
> Most efficient-reasoning baselines listed by the reviewer:
>
> * still rely on token-based CoT generation (long or short),
> * or rely on RL / preference optimization,
> * or require multi-sample rollouts or iteration,
> * or require significantly larger training datasets.
>
> In contrast, **LC-FT trains on only ~1k concise sketches**, requires no rollouts, no CoT generation during inference, and no reinforcement learning.
>
> For this reason, while conceptually related, many of these efficient-reasoning baselines operate in a *different inference paradigm* and were not selected as direct comparisons.
> We agree they are relevant and will enhance the related work discussion; however, we believe most of these methods would not be able to achieve meaningful improvements under the very small training-data regime used in LC-FT, which further differentiates our design setting.
>
> ---
>
> ## **2. Generalization to GPQA
>
> To directly test generalization, we evaluate LC-FT on **GPQA** using the model trained *only on concise OlympiadBench*:
>
> | Method             | Accuracy (%) | Avg Tokens  |
> | ------------------ | ------------ | ----------- |
> | Qwen3 Zero-shot    | 46.65        | 2672.49     |
> | LoRA (Q→A only)    | 47.99        | 2912.69     |
> | **LC-FT (Fast)**   | **49.11**    | **2763.46** |
> | Qwen3-Thinking     | 57.81        | 8631.43     |
> | **LC-FT + Router** | **55.80**    | **5118.02** |
>
> LC-FT improves over both zero-shot and LoRA despite having *no exposure* to STEM data, demonstrating that the learned strategy priors transfer beyond the training distribution.
>
> ---
>
> ## **3. “Framework complexity / training difficulty”**
>
> Although LC-FT consists of multiple conceptual components, it is lightweight in practice:
>
> * Stage 1 alignment (with frozen backbone) trains in **minutes** on a single RTX 4090;
> * Stage 2 SFT uses **only Q→A pairs**, identical to standard fine-tuning;
> * The codebook is small (a few latent vectors), not a secondary model;
> * Inference in fast mode uses *one forward pass* and answer-only decoding.
>
> Thus, despite its conceptual modularity, the actual training and inference pipeline is very accessible and requires minimal engineering overhead.
>
> ---
>
> ## **4. How LC-FT handles accuracy–efficiency trade-offs**
>
> The GainRouter (when enabled) relies on:
>
> * latent difficulty cues embedded in the thinking vectors, and
> * query representations of the problem itself.
>
> This combination allows the system to choose slow reasoning only when necessary.
> Our experiments—including GPQA—show that the router escalates selectively, preserving accuracy on difficult problems while avoiding unnecessary CoT on easier ones.
> On complex tasks, the mechanism naturally biases toward slow mode, matching the reviewer’s expectation.
>
> ---
>
> ## **5. Ablation: “Removing components does not significantly degrade performance”**
>
> The mild differences shown in the figure come from the plot scale and presentation.
> In fact, **removing the Codebook Layer leads to substantial degradation**.
>
> For example:
>
> * On **AIME**, removing the Codebook Layer causes the model to drop **below the baseline LoRA performance**,
> * and in some cases matches the “no-reasoning” baseline, indicating that the latent priors carry critical reasoning ability.
>
> Thus, the codebook is not a cosmetic component—it is central to LC-FT’s fast reasoning capability.
> ---
> We thank the reviewer again for the insightful comments.
> LC-FT differs fundamentally from existing efficient-reasoning approaches by performing reasoning entirely within the hidden space through a **single forward-pass**, without generating or iterating over CoT tokens.
> We hope the clarifications above, along with new out-of-domain evidence, help address the reviewer’s concerns.

---

> > ### Comment · Reviewer_G29z · 2025-11-22
> >
> > Regarding Q1, I believe these methods, including SFT, latent space intervention, and prompting, are highly effective approaches for efficient reasoning. **I think the authors should not only mention them in the Introduction and Related Work but also supplement relevant experiments in the Experiment section.** This would enhance the comprehensiveness and rigor of the paper. In the main text, the authors only compared zero-shot, few-shot, LoRA, and SoftCoT, which are **relatively outdated baselines.** Relying solely on these does not adequately demonstrate the effectiveness of the proposed method.
> >
> > Regarding Q2, **why did the authors not evaluate AIME24 and AIME25 separately**? Moreover, all evaluations were conducted only once, which introduces **significant randomness**. As seen in DeepSeek-R1 [1], they evaluated each problem 32 times and took the average as the final result. Additionally, since this method involves a **trade-off between performance and length**, it would be appropriate to reference existing literature and adopt a similar metric。
> >
> > [1] DeepSeek-R1 incentivizes reasoning in LLMs through reinforcement learning, Nature.
> >
> > Regarding Q3, this paper includes several complex modules that require meticulous hyperparameter tuning, which limits its **generalizability**. The authors need to provide a more detailed explanation of this aspect.

---

### Official Review · Reviewer_RYHy · 2025-10-27

**Soundness:** 1
**Presentation:** 1
**Contribution:** 2
**Rating:** 2
**Confidence:** 4

**Summary:**

This paper uses distillation and a codebook mechanism to enable concise reasoning, along with a router so that it can have longer thinking for more difficult questions.

**Strengths:**

- The motivation is plausible, as adaptive thinking based on question difficulty is promising.
- Simple routing actually increases the performance on math and code.

**Weaknesses:**

- There are many papers on efficient reasoning other than just routing papers (e.g., rejection fine-tuning, concise reinforcement learning with length penalty, etc). This paper totally ignores all of these.
- Fundamentally, this is distillation, as the model is trained to imitate the concise reasoning of teacher models. The codebook is just an abstract representation to use. Also, the codebook is not generally applicable, as the codebook and LLM are trained simultaneously. I think the introduction and overall paper fail to outline the core idea.
- Also, why don’t you compare with full fine-tuning of distillation training, as that would make a fair comparison?
- Also, the router is too naive and heuristic, raising concerns about applicability and generalization. Moreover, the gains mostly come from the router, so I am highly skeptical of why that complex codebook training is required.
- The method also requires two separate models, causing memory issues.
- Confusing terms and claims:
    - Fast and Slow Thinking
    - What it means by “Strong multi-step reasoning in practice often relies on either process supervision or preference-based policy optimization,” as recent reasoning models often rely only on outcome-based reward.
    - “Latent prior learning” is somewhat ambiguous.

**Questions:**

- What exactly is a hint r in Section 3.1?
- Please clarify what is “CoT Model” in Figure 2.
- For baseline “lora”, what data do you use?
- Why are there some tokens before “Answer” in Answer tokens?
- Some math operators look weird, like *raw_logit*.

---

> ### Author Response · Authors · 2025-11-17
> **Rebuttal to Reviewer RYHy**
>
> ### Rebuttal to Reviewer RYHy
>
> We thank the reviewer for the careful reading and constructive comments. We address each point below and hope these clarifications help convey the intent and contributions of our work.
>
> ---
>
> ## **1. “This is just distillation.” — LC-FT uses a different reasoning paradigm**
>
> LC-FT is not conventional distillation. Its key contribution is a **single-forward hidden-space reasoning mechanism**:
> the model retrieves K latent “thinking vectors” inside one forward pass, generates **no rationale tokens**, performs **no iterative refinement**, and then decodes **only the final answer**, just like non-thinking mode.
>
> Thus, LC-FT performs reasoning inside hidden representations rather than through explicit CoT generation or latent multi-step updates.
>
> ---
>
> ## **2. “The codebook is not generally applicable.” — Out-of-domain evidence**
>
> The codebook is small, trained only during Stage 1 with the backbone frozen, and encodes **abstract strategy heuristics** rather than problem-specific traces.
>
> To test generality, we evaluated the model trained *only on concise OlympiadBench* on GPQA:
>
> | Method             | Accuracy (%) | Avg Tokens  |
> | ------------------ | ------------ | ----------- |
> | Qwen3 Zero-shot    | 46.65        | 2672.49     |
> | LoRA (Q→A only)    | 47.99        | 2912.69     |
> | **LC-FT (Fast)**   | **49.11**    | **2763.46** |
> | Qwen3-Thinking     | 57.81        | 8631.43     |
> | **LC-FT + Router** | **55.80**    | **5118.02** |
>
> Fast LC-FT improves over zero-shot and LoRA without any STEM training, indicating that the learned priors transfer beyond the training domain.
>
> ---
>
> ## **3. “Most gains come from the router; why is the codebook needed?”**
>
> Two observations clarify this:
>
> * **Fast LC-FT (without router)** already outperforms zero-shot and LoRA, showing that the latent priors themselves are beneficial.
> * The router is intentionally simple; its effectiveness relies on the **difficulty and uncertainty cues encoded in the thinking vectors**, rather than on the router’s complexity.
>
> Thus, the codebook and router address complementary aspects.
>
> ---
>
> ## **4. “Why no comparison with full distillation?”**
>
> Full CoT-distillation requires training the model to generate teacher rationales.
> This contradicts LC-FT’s aim: inference should remain **CoT-free**, with reasoning carried out in hidden space.
> Hence, answer-only SFT and LoRA—matching LC-FT’s inference behavior—are the appropriate baselines.
>
> ---
>
> ## **5. “Two models cause memory issues.” — More precise clarification**
>
> Thank you for raising this point.
> The dual-model setup was an **engineering workaround**, not a limitation of LC-FT.
>
> ### **In principle**
>
> A single hybrid model (e.g., Qwen3-Base) can support both fast and slow modes via the `<think></think>` switch.
>
> ### **In practice**
>
> During our fine-tuning, **Qwen3-Base unexpectedly triggered tool-calling behavior**, even when tool usage was not desired.
> To ensure stable slow-CoT behavior, we used the Qwen3 series’ **subsequently released separate checkpoints**:
>
> * **Qwen3-Instruct** (non-thinking)
> * **Qwen3-Thinking** (reasoning mode)
>
> Thus, although two models were loaded for stability, LC-FT itself is fully compatible with a single unified checkpoint.
>
> ---
>
> # **Responses to Clarification Questions**
>
> ### **Q1. What is a hint (r)?**
>
> A concise, **non-empty**, high-level strategy cue generated by the teacher.
> It must be non-leaking and helpful, providing abstract heuristics rather than full reasoning chains.
>
> ---
>
> ### **Q2. What is the “CoT Model” in Fig. 2?**
>
> It refers to the **teacher model** used to generate and verify concise hints during data construction.
>
> ---
>
> ### **Q3. What data is used for the LoRA baseline?**
>
> LoRA is trained on **exactly the same Q→A pairs** used in LC-FT’s Stage 2 SFT.
> No hints or CoT sequences are used for LoRA.
>
> To clarify the symmetry:
>
> * LC-FT **Stage 1** uses hints only to align thinking-token slots.
> * LC-FT **Stage 2**, like LoRA, uses **only answer-token supervision**.
>
> Thus, the comparison is fair.
>
> ---
>
> ### **Q4. Why are tokens shown before “Answer”?**
>
> These come from the backbone model’s default instruction preamble; LC-FT does not introduce additional tokens.
>
> ---
>
> ### **Q5. Some math operators look unusual.**
>
> Thank you for pointing this out. We will refine the mathematical notation to ensure clearer and more consistent expression.
>
> ---
>
> We appreciate the reviewer’s thoughtful feedback.
> We hope the above clarifications and evidence address the reviewer’s questions.

---

### Official Review · Reviewer_eamC · 2025-10-28

**Soundness:** 3
**Presentation:** 2
**Contribution:** 2
**Rating:** 4
**Confidence:** 3

**Summary:**

The authors propose "Latent Codebooks for Fast Thinking", a learned latent codebook of discrete, reusable priors.

To train this approach, the authors first build a dataset of concise CoTs, by prompting a teacher model to generate "hints" from $x$ and $y$. This hint is accepted i.f.f. no answer leakage is detected, and conditioning on the hint leads the student to a successful answer (these hints seem analogous to a strategy for solving the problem at hand).

The codebook is integrated via cross attention, via a learnable query matrix that's shared across inputs.
At a given layer, the input is augmented on the sequence length with $K$ thinking tokens, concatenated to the right, so they can attend the previous input. A separate "Refiner" MLP additionally processes the thinking tokens.

The actual training is in two phases : in the alignment phase, a frozen LLM processes $(x, r)$, where $r$'s representations are mean-pooled at layers $l >L$, which will serve as alignment targets for the mean-pooled thinking token slots of the same layer. This can be seen as distilling the rationales inside the thinking tokens. In the Supervised Fine-Tuning phase, the teacher model and the rationales are dropped, and the new architecture is trained to predict directly from the input and the thinking tokens the answer.

To account for settings where a slower, lengthier generation would be better, the authors propose the GainRouter,  a lightweight router switching between this "condensed" mode of reasoning, and "standard" reasoning. Its input is a function for both the pooled query tokens and the thinking tokens, trained with BCE loss on whether only the slow model succeeds.

Experiments are conducted with the Qwen base model, and compared against small set of baselines on math and coding benchmarks. Finally the authors provide an ablation study on several components of their proposed methods, showing the effectiveness of the codebook.

**Strengths:**

1. The authors tackle an important question of efficiency in reasoning that is plaguing LLMs at test-time. The latent embedding approach is novel and interesting.

2. The authors provide an extensive ablation on some of the different components of their approach.

**Weaknesses:**

1. The proposed approach is somewhat convoluted; a separate data creation step, a learned codebook, additional cross-attention, two-phase training, an added refiner MLP, several losses, and an additional router. After reading the paper, I am unsure I truly have enough of the implementation details to  exactly implement the approach. Given that the authors have shared code, I am ok with overlooking the last point (that being said the link doesn't work so please fix it). That being said, overall given the complexity of the approach, it's unclear to me whether the small but somewhat consistent gains makes this a viable option for practical use cases.

2. The results (specifically figure 4 left) for trading off performance would be better represented as Pareto curves; the authors could simply change the threshold in the GainRouter to use or not full reasoning; this way we could see how performance is "interpolated" across Slow and Fast Training.

**Questions:**

Questions :
1. "Moreover, strong multi-step reasoning in practice often relies on either process supervision or preferencebased policy optimization which raises data and engineering barriers for large-scale deployment Schulman et al. (2017)"
After reading the paper it's unclear how your approach bypasses this; to generate your training data you are still going through the process of generating multiple rollouts and keeping the successful ones, which is the de-facto approach for CoT training.

2. Figure 2 is not clear; is the bottom left the generated hint ? is CoT model == Teacher Model ?
3. Figure 3 is a quite dense; what exactly is $L_q$ ? What is $S$ is Layer $S-1$ ? Why does it say alignment loss $L_{LM}$ on the answer tokens in the middle figure, are you calling the next-word prediction objective an alignment loss, or is this something else ?

4. The authors phrase the training dataset as containing "Concise CoT", but this seems somewhat poorly named, given that the CoT don't contain the actual chain of though, but rather "hints", that could be incomplete. For example, if the base model can already solve the problem at hand,  $\hat{r}$ could be empty, hence pass verification, but is not a CoT.

4. If you are using cross-attention (and not self-attention), why do you need to pad your input ? Why not just process without padding for the first L layers, and then concatenate the thinking tokens ? This should be doable if your sequences are left padded.

Comments :
1. please use `\citep{}` in latex when the citation is not part of the sentence's syntax or meaning.

---

> ### Author Response · Authors · 2025-11-17
> **Rebuttal to Reviewer eamC**
>
> ### Rebuttal to Reviewer eamC
>
> We thank the reviewer for the detailed and constructive feedback. We respond to each weakness and question below.
>
> ---
>
> ## **Weakness 1: “The approach appears convoluted.”**
>
> We understand this concern, and part of the impression may have come from our presentation.
> First, **we apologize for the earlier issue with the anonymous code link—it has now been fixed**.
>
> In practice, the implementation is lightweight:
>
> * Training on OlympiadBench (2 epochs alignment + 3 epochs SFT) finishes in **under 10 minutes on a single RTX 4090**.
> * Codebook retrieval is a **single cross-attention block**;
> * The Refiner is a **small MLP applied only to K hidden slots**;
> * No RL, no multi-rollout filtering, no iterative refinement.
>
> Ablations show the codebook contributes the majority of the performance gains. We will simplify the description in future versions to better reflect this simplicity.
>
> ---
>
> ## **Weakness 2: “Performance trade-off should be shown as Pareto curves.”**
>
> Thank you for the suggestion.
> Our current GainRouter uses a **learnable gating head**, which implicitly learns the switching threshold.
> We agree that sweeping fixed thresholds to produce explicit accuracy–token curves would further illuminate the trade-off, and we will explore adding these analyses in future versions.
>
> ---
>
> # **Responses to Questions**
>
> ---
>
> ## **Q1. “Unclear how the method bypasses process supervision if concise CoT still requires rollouts.”**
>
> Our claim concerns avoiding *large-scale* or *inference-time* process supervision.
> In LC-FT:
>
> * Concise hints are generated **once, offline**, on a small dataset;
> * No iterative exploration or RL is used;
> * At inference, LC-FT **does not generate CoT**, only the final answer.
>
> Thus, the method avoids both the major computational cost and the inference overhead typical of process-supervised approaches.
>
> ---
>
> ## **Q2. “Figure 2 is unclear.”**
>
> Yes—the bottom-left text is the **concise hint** generated by the teacher, and “CoT model” refers to the teacher model. We will revise the figure in future versions.
>
> ---
>
> ## **Q3. “Figure 3: What is L₂? What is S? Why alignment loss on answer tokens?”**
>
> Thank you for pointing this out:
>
> * (L_2) denotes the **starting alignment layer**, corresponding to the insertion layer (L).
> * (S = {L, L+1, ...}) is the set of layers where we align the thinking-token slots with the teacher’s hint representations.
> * The label “alignment loss on answer tokens” is a **figure error**; this term refers to the **SFT loss** applied to answer tokens during Stage 2.
>
> We will correct these issues in later versions.
>
> ---
>
> ## **Q4. “Concise CoT may be incomplete; r̂ may be empty.”**
>
> To clarify: **hints are never empty** in our data pipeline.
> Even when the base model can solve the problem, the teacher is required to produce a **minimal but non-empty** concise hint that passes verification (non-leaking, helpful).
>
> ### **Why concise hints are beneficial and generalizable**
>
> These hints are **strategy-level cues**, not full reasoning traces. This offers key advantages:
>
> * **Better generalization:** hints encode abstract reasoning heuristics (e.g., inspect parity, consider boundary cases), not long instance-specific steps.
> * **Lower numeric sensitivity:** hints are robust to value changes because they avoid detailed arithmetic chains.
> * **Cleaner supervision:** concise sketches reduce noise common in long CoT.
> * **Reusable priors:** these high-level signals support the codebook in learning *transferable* strategy priors, which helps explain LC-FT’s out-of-domain performance (e.g., on GPQA).
>
> ---
>
> ## **Q5. “Why pad K thinking tokens if using cross-attention?”**
>
> Padding simplifies and stabilizes the design:
>
> * Tensor shapes remain fixed across layers;
> * Masking is straightforward (slots masked before layer (L), visible after);
> * Refiner and downstream blocks rely on consistent positional anchors;
> * Avoids the significant engineering complexity of dynamically changing sequence length mid-forward pass.
>
> This choice keeps the implementation robust and efficient.
>
> ---
>
> ## **Q6. “Use \citep when citation is not part of the sentence.”**
>
> Thank you—we will fix citation formatting in future revisions.
>
> ---
>
> We appreciate the reviewer’s careful reading and helpful suggestions.
> Several concerns stemmed from unclear exposition, and we will refine the relevant sections and figures in future versions to improve clarity.
> Thank you again for your constructive feedback.

---

### Official Review · Reviewer_6hoU · 2025-11-01

**Soundness:** 2
**Presentation:** 2
**Contribution:** 2
**Rating:** 4
**Confidence:** 3

**Summary:**

The authors introduce Latent Codebooks for Fast Thinking (LC-FT), a framework that uses concise CoT sketches during training to learn a latent codebook of reusable strategy priors.
During inference, LC-FT prepend K continuous “thinking vectors” generated from the codebook before CoT generation.
It also has a GainRouter module that adaptively decides when to use fast (codebook-guided) vs. slow (explicit CoT) reasoning, balancing accuracy and efficiency.

**Strengths:**

1. The idea of storing soft representation of reasoning strategies for retrieval at inference time is novel.
2. The use of codebook seems to improve model performance compared to vanilla SFT as shown in table 1 and 2.
3. The router significantly improves the performance, while also significantly increase generation length.

**Weaknesses:**

1. The writing is a bit unclear, especially the method section. It took me a while to understand that the K soft thinking vectors are prepend to the CoT instead of replacing the CoT. Maybe make that clear at the begining.
2. The paper is about "fast thinking", but the thinking speed up seem to purely coming from the teacher generated concise CoT. The method itself actually adds K soft tokens in front of the CoT. As shown in Table 1 and 2, the generation length of LC-FT is roughly the same as SFT on the concise CoTs. The generation is even significantly longer than base model when using a router on top of LC-FT. To me the paper is closer to a parameter-efficient training method for reasoning tasks instead of "fast thinking".
3. The two training sets are quite small (1k+) and only four testing sets are considered. I wonder how generalizable is the proposed method. Would the learned codebook only cover a specific set of problems and strategies so the method can only be applied to one narrow domain?
4. typos: mis-formated equations on line 254

**Questions:**

See weeknesses.

---

> ### Author Response · Authors · 2025-11-17
> **Rebuttal to Reviewer 6hoU**
>
> ### Rebuttal to Reviewer 6hoU
>
> We sincerely thank the reviewer for the careful reading and constructive feedback. We address each of your points in turn.
>
> ---
>
> #### 1. “The writing is a bit unclear… K soft thinking vectors are prepend to the CoT instead of replacing the CoT.”
>
> Thank you for pointing this out. Our exposition was indeed not sufficiently clear and led to this misunderstanding.
>
> In LC-FT, the **K thinking vectors are never prepended to a CoT sequence**, and in fast mode there is in fact **no CoT generated at all**:
>
> * The thinking vectors are **hidden-state slots** injected once at an internal transformer layer during the prefill stage.
> * They **do not correspond to textual tokens** and are **never decoded**.
> * In fast mode, the model directly outputs the **answer-only**, without any chain-of-thought decoding.
>
> In other words, the thinking vectors only affect the *internal computation* of the model, not the surface form of the output. We agree that this should have been made explicit earlier in the Method section, and will revise the text accordingly to avoid the impression that we prepend K tokens before a CoT.
>
> ---
>
> #### 2. “The paper is about ‘fast thinking’, but the speed up seems to purely come from the teacher generated concise CoT… the method itself actually adds K soft tokens in front of the CoT… generation length… router…”
>
> We appreciate this concern about what exactly we mean by “fast thinking”.
>
> Our intention is that **fast thinking refers to where reasoning happens**:
>
> * Existing CoT-style methods perform reasoning **in token space**, by explicitly generating multi-step rationales.
> * LC-FT performs reasoning **in hidden space**: a latent codebook produces K thinking vectors that are injected once inside the network to provide **strategy-level priors**, and the model then **decodes only the final answer**.
>
> The **concise CoT sketches are used only during training** to align the latent codebook; they are never decoded at inference. The K thinking vectors are not output tokens and do not affect sequence length. This is why, as you observed, the generation length of LC-FT in fast mode is similar to answer-only SFT—this is by design: we do not generate any reasoning tokens.
>
> Regarding the router: it indeed increases generation length, but only when it intentionally switches to **slow, explicit CoT reasoning** for harder questions. For easier questions it stays in fast mode and keeps the decoding short. This behavior reflects the intended trade-off between efficiency and accuracy, rather than an unintended overhead.
>
> We agree that, in this sense, LC-FT can also be viewed as a parameter-efficient way to endow a base model with an additional “fast reasoning” capability; we will clarify the terminology around “fast thinking” to make this perspective clearer.
>
> ---
>
> #### 3. “The two training sets are quite small… I wonder how generalizable is the proposed method… narrow domain?”
>
> This is an important concern. To directly assess generalization beyond the relatively small Olympiad-style training data, we evaluated LC-FT on **GPQA**, a challenging STEM reasoning benchmark that is **out-of-domain** relative to our training distribution.
>
> > Importantly, for this experiment we use **exactly the LC-FT model trained only on concise OlympiadBench data** (no GPQA, no STEM-specific finetuning or adaptation of any kind).
>
> The out-of-domain results are:
>
> **GPQA (out-of-domain, zero-shot):**
>
> | Method             | Accuracy (%) | Avg Tokens  |
> | ------------------ | ------------ | ----------- |
> | Qwen3 Zero-shot    | 46.65        | 2672.49     |
> | LoRA (Q→A only)    | 47.99        | 2912.69     |
> | **LC-FT (Fast)**   | **49.11**    | **2763.46** |
> | Qwen3-Thinking     | 57.81        | 8631.43     |
> | **LC-FT + Router** | **55.80**    | **5118.02** |
>
> Fast LC-FT, without any GPQA training, already outperforms both zero-shot and LoRA baselines. The router mode further approaches the performance of the full CoT model with substantially fewer tokens. This suggests that the latent codebook is capturing **reusable strategy priors** that transfer beyond the specific problems seen in the small Olympiad training set, rather than overfitting to a narrow domain.
>
> We acknowledge that our current experiments cover only four benchmarks; we will explicitly mention this as a limitation and plan to evaluate on more diverse datasets in future work.
>
> ---
>
> #### 4. “Typos: mis-formatted equations on line 254”
>
> We appreciate you pointing this out. We have identified the misformatted equations and will correct them, along with other minor typographical issues, in the revised version.
>
> ---
>
> Once again, we thank you for your careful reading. Several of your comments stem from places where our exposition was not sufficiently precise, and we apologize for that. We will substantially improve the description of the inference procedure, the role of thinking vectors, and our definition of “fast thinking” in the camera-ready version.

---

### Meta-Review · Area_Chair_GKX8 · 2026-01-07

**Summary:**

1. Inadequate baseline coverage. Need to compare with more recent techniques
2. Missing detailed descriptions about the major concepts proposed in the work. E.g., how is fast/slow thinking proposed in this paper related to recent popular fast/slow/hybrid thinking paradigms.
3. The proposed method is quite complicated. Hard to justify the necessarity of each step.
4. Experiments are relatively small scale.

**Reviewer Concerns:**

Most of the concerns are only partially addressed in the rebuttal (e.g., generalize to GPQA). Further reviewer's question is not addressed (G29z).

**Reviewer Scores:**

Given that the rebuttal doesn't address the issues fully, reviewers likely will not change their scores (4442).

---

### Decision · Program_Chairs · 2026-01-26

Reject